# Antiviral Plants from Marajó Island, Brazilian Amazon: A Narrative Review

**DOI:** 10.3390/molecules27051542

**Published:** 2022-02-24

**Authors:** Paulo Wender P. Gomes, Luiza Martins, Emilli Gomes, Abraão Muribeca, Sônia Pamplona, Andrea Komesu, Carissa Bichara, Mahendra Rai, Consuelo Silva, Milton Silva

**Affiliations:** 1Collaborative Mass Spectrometry Innovation Center, Skaggs School of Pharmacy and Pharmaceutical Sciences, University of California San Diego, San Diego, CA 92122, USA; wendergomes@health.ucsd.edu; 2Laboratory of Liquid Chromatography (Labcrol), Institute of Exact and Natural Sciences, Federal University of Pará, Belem 66075-110, Brazil; abraao_muribeca@hotmail.com (A.M.); sgpamplona@yahoo.com.br (S.P.); carissa.bichara@ufra.edu.br (C.B.); yumikoyoshioka@yahoo.com.br (C.S.); 3Institute of Animal Health and Production (ISPA), Federal Rural University of Amazonia (UFRA), Av. Presidente Tancredo Neves No. 2501, Terra Firme, Belem 66077-830, Brazil; luiza.martins@ufra.edu.br; 4Institute of Health Sciences, Faculty of Nutrition (FANUT), Federal University of Pará (UFPA), R. Augusto Corrêa, 01, Guamá, Belem 66075-110, Brazil; emillirsousa@gmail.com; 5Department of Marine Sciences (DCMar), Federal University of São Paulo (UNIFESP), R. Carvalho de Mendonça, 144, Encruzilhada, Santos 11070-100, Brazil; andrea_komesu@hotmail.com; 6Biotechnology Department, SGB Amravati University, Amravati 444 602, India; mahendrarai7@gmail.com; 7Faculty of Pharmaceutical Sciences, Federal University of Pará, Belem 66075-110, Brazil

**Keywords:** viral diseases, plants extract, medicinal plants, Amazon region

## Abstract

Diseases caused by viruses are a global threat, resulting in serious medical and social problems for humanity. They are the main contributors to many minor and major outbreaks, epidemics, and pandemics worldwide. Over the years, medicinal plants have been used as a complementary treatment in a range of diseases. In this sense, this review addresses promising antiviral plants from Marajó island, a part of the Amazon region, which is known to present a very wide biodiversity of medicinal plants. The present review has been limited to articles and abstracts available in Scopus, Web of Science, Science Direct, Scielo, PubMed, and Google Scholar, as well as the patent offices in Brazil (INPI), United States (USPTO), Europe (EPO) and World Intellectual Property Organization (WIPO). As a result, some plants from Marajó island were reported to have actions against HIV-1,2, HSV-1,2, SARS-CoV-2, HAV and HBV, Poliovirus, and influenza. Our major conclusion is that plants of the Marajó region show promising perspectives regarding pharmacological potential in combatting future viral diseases.

## 1. Introduction

Viruses are a global threat and, in addition to health problems, cause serious social problems to humanity. They are the main contributors to many minor and major outbreaks, epidemics, and pandemics worldwide, such as swine flu, avian influenza, dengue fever, and the current SARS-CoV-2 that causes COVID-19 [1,2]. Although several treatment methods are available for viral diseases, viruses evolve rapidly and this requires a search for new potent and effective antiviral compounds with fewer or, better yet, no adverse side effects at all [1].

Medicinal plants have been documented as a means of complementary treatments that are useful for a range of diseases. However, few have been applied to the treatment of viral diseases. This is because some viral targets have very specific interactions with with natural molecules. Furthermore, most antivirals that are used as drugs show undesirable side effects, or the virus ends up developing a resistance to the drug, as well as recurrence and latency. In this sense, many medicinal plants can be candidates for the discovery of new antivirals due to their strong antiviral activity, fewer side effects, and the capacity to inhibit the replication or synthesis of some virus genome [3]. In addition, science has already proved the safer benefits of herbal medicines and also supports the idea of improved therapy when these herbal medicines are used in combination with conventional antiviral medicines [4].

Among many islands around the world, the island of Marajó which is part of the Brazilian Amazon, is known to have several cultures formed by indigenous, quilombolas, and mestizo populations that make use of traditional knowledge of plants, which are sometimes used as the only form of therapy in their traditional medicine. The Amazon region also has a very wide plant diversity, being estimated to have from 25 to 30 thousand species of endemic plants, and some species are also associated with the treatment of diseases [5]. In this sense, according to the literature [6], many plants from Marajó island are frequently used for the treatment of diseases, and could be a source of new specific bioactive compounds. This can be applied to expand promising studies.

Based on this, we highlight the importance of searching for new natural sources with antiviral potential, as they can be an alternative to high-cost synthetic drugs with undesirable side effects, in addition to facilitating access to the population. Therefore, we describe a brief narrative review of the main plants inserted in the ethnomedicinal use of Marajó island to treat a lot of diseases, which are directly or indirectly associated with viral agents.

## 2. Data Research Methodology

The present narrative review originated from the empirical knowledge of some authors of this research and the daily coexistence with the ethnomedical use of plants from the Marajó Island. Posteriorly, the pharmacological and botanical correspondence with the species and its reported therapeutic indication were conformed to specific search descriptors terms in Scopus, Web of Science, Science Direct, Scielo, PubMed, and Google Scholar, as well as the patent offices in Brazil (INPI), United States (USPTO), Europe (EPO) and World Intellectual Property Organization (WIPO). The main keywords of the search were “Marajó”, “antiviral”, “medicinal herbs”, “herbal medicines”, “medicinal plants”, and “traditional medicine”. We emphasize that, in addition to these terms, the names of the species of interest were used after filtering the twelve plants reported in this study.

## 3. Main Vectors of Diseases Caused by Pathogenic Viruses

The most emerging viruses are zoonotic, which means that they can be transmitted from animals to humans, especially from mammals of the orders Chiroptera, Rodentia, Primates, and Carnivores, as they are hosts to a large diversity of viruses (see Figure 1). For instance, the literature investigated bats as the main vectors responsible for the most zoonotic viral infections. However, the total number of viruses identified in rodents is higher than that in bats [7]. In any case, whenever a virus jumps from animal to human, society is at serious risk.

At present, the world is on alert to the emergence of viruses that are dangerous to public health and have the potential to begin a new pandemic. However, most zoonotic viruses transmitted from animals to humans are not easily transmitted between humans. In this sense, we can have a new problem: humans as new hosts [7].

In Brazil, other important vectors are mosquitoes and ticks, which transmit arbovirus to vertebrates, and are known as arthropod-borne viruses. Arthropods that facilitate efficient virus transmission between susceptible hosts are called vectors. Several arboviruses that are pathogenic to humans, such as the yellow fever virus (YFV), the Chikungunya virus (CHIKV), and the Zika virus (ZIKV), share the same vector, *Aedes aegypti* mosquito, which are abundant in tropical and subtropical regions [7].

## 4. The Main Antiviral Plants Used to Treat Viral Diseases in Marajó Island

Many factors motivated the search for new antiviral agents from medicinal plants. From these, we highlight the biggest limitations to the effectiveness of the drugs that are currently available on the market and, although there are several options for antiviral drugs with application in several existing viruses, many of them remain intractable with these drugs. In addition, many synthetic drugs can cause viral resistance and latency. In this sense, the knowledge and wisdom of the natives regarding medicinal plants continues to be a source for the discovery of new molecules with the potential for use as new or complementary antivirals in the treatment of diseases caused by viruses [8].

In the infield of research in the ethnopharmacological area on Marajó, we found reports of plants that have met the immediate needs of communities in this area. In this sense, interest in the diversity of the flora of the Marajó region is not recent, and has led to studies of prospective bioactive compounds from medicinal plants [9] with activity against viruses (Figure 2), such as those caused by Human Immunodeficiency Virus (HIV), Herpes Simplex Virus types 1 and 2 (HSV-1,2), Hepatitis A and B Virus (HAV/HBV), Poliovirus, influenza, and SARS-CoV-2. The influence of many people (Europeans, Africans, etc.), and the exchange of knowledge between such people, continues to reverberate, as can be seen through popular traditions of using medicinal plants [9].

In Table 1, we listed some plants used in Marajó to treat diseases caused by viruses. This was narrated by researchers from this group, who are also natives of the region (authors: P.W.P.G., E.G., and A.M.). In other places of the world, these plants are empirically used for other purposes. We also emphasize that all species have vernacular names, as well as an empirical therapeutic indication. As a complement to this, our group established scientific attributions for in vitro action against viruses already reported in the literature.

## 5. Antiviral Compounds of Medicinal Plants from Marajó

Plants in general naturally have defense mechanisms against microorganisms, especially viruses, which are low-molecular-weight secondary metabolites with a level of toxicity. This factor can explain why such plants have been used therapeutically for viral diseases. Based on a metric review of the literature, Table 2 summarizes the main compounds involved in the antiviral activity of the medicinal plants from Marajo.

### 5.1. Alpinia zerumbet *L*.

*Alpinia zerumbet* L. is a medicinal plant used in many states in the north and northeast of Brazil [10]. In the Amazon, it has the vernacular name of “Vindicá”. More specifically, in Marajó, the leaves are used to prepare tea and baths that empirically help or bring some benefit for the treatment of flu and colds. To corroborate these antiviral reports, aromatic heterocycles of compounds isolated from the leaves and rhizomes of *Alpinia zerumbet* showed excellent antiviral activity for human immunodeficiency virus type 1 by the mechanism of inhibition HIV-1 integrase and neuraminidase activity [11]. Moreover, extracts from this species containing dehidrokavaina (DK), and dihydro-5,6-dehydrokawain (DDK) showed the same activity (see Figure 2), with IC_50_ of values of 30 and 188 μg/mL. These data confirm and highlight the antiviral potential of that species, which is well distributed in the Marajó island.

### 5.2. Bixa Orellana

Another plant with antiviral properties is *Bixa Orellana*, also named “Urucum”. It is a very common plant in the northern region of Brazil; however it is also present in other countries of South America, such as Peru, Columbia, Ecuador and Mexico [12,13]. Two antiviral compounds have been isolated from this plant: Procyanidin B2 and Lutein. These compounds block the viral binding to the cell receptors for influenza and inhibit the HBV protease. In this sense, *B. orellana* is a promising source of compounds that could be applied in antiviral therapy [14,15,16].

**Table 2 molecules-27-01542-t002:** Chemical components are probably involved in antiviral activities for main medicinal plants from Marajó Island.

Medicinal Plant	Classes Chemical	Assignment Viral	Antiviral Agents	Methods	Action Mechanism	IC_50_ (µg/mL)	References
*Alpinia zerumbet* L.	Aromatic heterocycles	HIV-1	5,6-Dehidrokavaina (DK)	Multiple integration assay	Inhibition HIV-1 integrase and Neuraminidase Activity	4.4 μg/mL	[11]
			Dihydro-5,6-dehydrokawain (DDK)			3.6 μg/mL	
*Bixa orellana* L.	Polyphenols	Influenza virus and hepatitis B virus	procyanidin B2	CPE inhibition assay	Blockage of viral binding to the cell receptors	50.85–56.02 μg/mL	[14,15,16]
			lutein	ELISA assay	Inhibition of HBV transcription	40 μg/mL	
*Citrus limon*	Terpenes	Hepatitis A virus (HAV)	Limonene, β-pinene, andγ-terpinene Limonexic acid	Reed and Muench method	Reducing HAV infectivity	2.84 log TCID_50_/mL	[17,18]
*Citrus limonum* Risso	Flavonoids	Hepatitis A Virus (HAV)	procyanidin B2	Reed and Muench method	The slight reduction in virus infectivity	2.84 log TCID_50_/mL	[17]
*Dysphania ambrosioides*( L.)	Flavonoids	SARS-CoV-2	rutin and nicotiflorin	3CLpro and RdRp	Molecular docking with 3CLpro (main protease (Mpro))	*in silico*	[19,20,21,22]
*Libidibia ferrea* (Mart. ex Tul.) L.P.Queiroz	Polysaccharide	HSV-1	Sulfated polysaccharide	Plaque reduction assay	Antiviral activity by its ability to prevent viral replication	405 μg/mL	[23]
		Poliovirus (PV)				1.25-10 μg/mL	
*Ocimum gratissimum* L.	Terpenes and Alkaloid	HSV-1,2	Eugenol	Plaque-based assays (PFU) method	Inhibition of virus replication	16.2, and 250 µg/mL	[24,25]
		HIV-1,2	Thymol, and ursolic acidPheophytin-a		Direct destruction of the virion	7 µM	
*Plectranthus amboinicus* (Lour.) Spreng	Flavonoids	HSV-1	Flavonoids	cleavage of peptide substrate	HIV-1 protease inhibitor	100 µg/mL	[26]
*Spondias mombin* L.	Flavonoids and phenolic acids	HSV-1	tannin-rich fraction	Vero E6 cells	Glycoproteins gB and gD of HSV-1 surface	17.35 µg/mL	[27]
			geraniin			20.40 µg/mL	

### 5.3. Citrus limon

The *Citrus limon* species is known in the Marajó island as “Limãozinho”. It has many pharmacological properties as well as important natural chemical constituents, including citric acid and ascorbic acid [28]. For some time, the pharmacological properties of this species were associated with the presence of vitamin C. However, recent data confirm the participation of other supporting substances [28], such as Terpenes [18], Limonene, β-pinene, and γ-terpinene, which have been associated with reductions in hepatitis A virus (HAV) infectivity [17] by reducing HAV infectivity (See Table 2). These results reinforce the potential of this species and place it in the window of natural products with antiviral activities.

### 5.4. Citrus limonum Risso

*Citrus limonum* Risso is commonly called “Limoeiro” in the Marajó Island [29]. It belongs to the Rutaceae family and has round and acidic fruits called lemon. Tea from the peels of *Citrus limonum* fruits is used as a natural expectorant, which helps in the treatment of flu and viruses that cause the accumulation of secretions in the lungs. Recently, the literature reviewed the essential oils of *Citrus limonum*., and reported several therapeutic benefits, including antiviral activity [30]. To corroborate its empirical use in the Marajó, the literature has already highlighted a role that proves its activity as an expectorant agent [31]. Sheppard and Boyd observed that the expectorant property of lemon oil, when inhaled, is mainly due to limonene (see Figure 2). Recent data reported its activity against influenza viruses [32,33], as well as activity against Hepatitis A Virus (HAV), attributed to Proanthocyanidins, by a slight reduction in virus infectivity [17].

### 5.5. Dysphania ambrosioides

*Dysphania ambrosioides*, also known as “mastruz”, is widely used in Brazil as a medicinal plant. It has been reported to have beneficial effects against respiratory diseases. The literature [19] carried out a study with a computational approach (in silico), evaluating the potential of the compounds present in this species, and found that such compounds could inhibit the particles involved in the replication of the SARS-CoV-2 virus, responsible for causing SARS-CoV-2. The focus was on the flavonoid and derivative compounds that are present in this species. The results suggest that the substances Rutin and Nicotiflorin, two of the main “mastruz” flavonoids, and their compounds showed promising potential to block 3CLpro and RdRp proteins, and could play a key role in decreasing or inactivating SARS-CoV-2 infection [19,20,21,22]. The study points to Rutin as a possible alternative to low-molecular-weight heparin (LMWH), due to its anticoagulant and anti-inflammatory effects and its potential protection against acute lung injury (ALI). A computational study has its limitations. Therefore, it is necessary to conduct more in-depth studies, in vitro, in vivo, and clinical, to consolidate the use of these flavonoids or other detected substances, such as medicine.

### 5.6. Libidibia ferrea

*Libidibia ferrea* has been suggested to inhibit the replication of HSV-1 and PV viruses. One pioneer study [23] reported the antiviral effect of *L.*
*ferrea*, and the authors evaluated the effect of the sulfated polysaccharide obtained from the aqueous extract of *L. ferrea* seeds and its antiviral activity, as well as its ability to prevent viral replication, and viral fixation, and its direct effect on viral particles. The sulfated polysaccharide from *L. ferrea* extract has a polyanionic character, as sulfated polysaccharides are potent inhibitors of HSV binding to host cells competing with viral glycoprotein receptors. Thus, it is assumed that these substances end up solubilizing the virus envelope, which can also cause chemical changes and the inhibition of essential proteins reaching the virus envelope [23]. The same inhibition effect of this sulfated *L. ferrea* polysaccharide was observed for Poliovirus (PV), and was even stronger in this case, as it interfered in the initial stages of virus replication. Thus, the study shows that this sulfated *L. ferrea* polysaccharide has potential as an antiviral, and these mechanisms are likely results of the action of the complex compounds in the extract composition, which may all act together to avoid virus replication [23].

### 5.7. Ocimum gratissimum *L*.

Plants of the genus *Ocimum* are widely used in African countries to treat HIV infection. The literature reported that *Ocimum gratissimum* leaf extract had 90% effective HIV inhibition after 2 h of infection, a better result than that obtained for AZT, a drug used in the treatment of SIDA, under the same conditions [25]. These results were attributed to Eugenol, Thymol, and ursolic acid caused by the inhibition of virus replication, and direct destruction of the virion mechanism. Moreover, a study conducted with *Ocimum* species reported activity of this species against Herpes Simplex Virus Type-1 (HSV-1) [34]. All pieces of evidence confirm the antiviral potential of *O. gratissimum,* as well as the antiviral reports by people from Marajó.

### 5.8. Plectranthus amboinicus

A flavonoid-rich fraction of the *Plectranthus amboinicus* showed antiviral activity against HIV-1 and the Herpes Simplex Virus type 1 [26]. It was checked for antiviral activity using the cleavage of peptide substrate method. It showed excellent antiviral activity in cell analysis, with IC_50_ 100 µg/mL, and mechanisms of action for inhibiting HIV-1 protease, showing potential for use as an antiviral agent.

### 5.9. Spondias mombin *L*.

*Spondias mombin* L. is a medicinal plant known vernacularly as “taperebazeiro”, which belongs to the Anacardiaceae family and is found in almost all Brazilian territory (except in the South). This species has been reported to have antiviral activity against Herpes Simplex Virus type 1 (HSV-1) [27]. This study demonstrated that a fraction enriched with tannins and mainly geraniin showed promising anti-HSV-1 activity. The replication cycle of HSV is widespread in molecular biology (1. viral adhesion; 2. entry into host cells; 3. genome modification; 4. replication of genetic material; 5. creation of new capsids; 6. propagation of new viral particles). In this sense, data in the literature highlight that the natural products of split *S. mombin* may have their main effects in the early stages of HSV-1 infection, and show a better immune response when incubated with viral particles (glycoproteins gB and gD of HSV-1 surface). We understand that in vivo and clinical studies in humans are needed to validate the properties reported here. However, we emphasize that these results are unique and reinforce the herbal potential of *S. mombin* to treat HSV-1 infections.

## 6. Future Perspectives

Some antiviral plants described here are also found in other regions, and some compounds can be detected in other species. However, the study of plants that are on Marajó island, although some are not native to the island, are fundamental for the inventory of active biological sources available in the region. Furthermore, scientific reports about those species could promote the development of Marajó, as the trade in medicinal plants could alleviate the poverty of the native peoples, and lead to significant independence, albeit small, from synthetic, expensive, and inaccessible medications. Therefore, these plants are promissory and can open prospects for the use of unexplored resources, which could help in the economic and sustainable use of Brazil’s tropical forests.

Based on the experimental designs available in the literature, we suggest a workflow (Figure 3) for the prospective and respective applications of antiviral candidates (e.g., Phenolics, alkaloids, and terpenes) from plants of the Marajó Island. We believe that studies based on this perspective may lead to promising and indispensable therapeutic alternatives. We carefully highlight, that for highly skilled researchers, this experimental design is already known. However, we also hope to instigate young researchers who carry a larger folk knowledge, for it is of paramount importance that the population receives a return on any resources generated from biodiversity. For this, comprehensive research needs to be developed and validated by regulatory agencies, to ensure that the population enjoys any benefits that these studies may generate. Furthermore, we highlight the importance of translating the folk knowledge of native people to the analytical metrics of scientific research, so that studies can be guided by unequivocal interests.

## 7. Conclusions

This study shows that the Marajó island (Amazon region), has a high diversity of medicinal plants with high antiviral potential, and must be further studied and valorized. It was seen that secondary metabolites from these species have bioactive compounds that can be useful if isolated and synthesized as potential antiviral agents; however, it is expected that such compounds should exhibit greater potency, selectivity, duration of action, bioavailability, and reduced toxicity. It is important to have a base in the traditional and scientific knowledge of the Marajó plants, as the plant kingdom is becoming an important source of new agents with special biological targets. In recent years, the efforts of several researchers have helped to discover and isolate many secondary metabolites from plants that are also found in Marajó island, have potential antiviral action and can be used in therapy against infectious diseases caused by viruses.

## Figures and Tables

**Figure 1 molecules-27-01542-f001:**
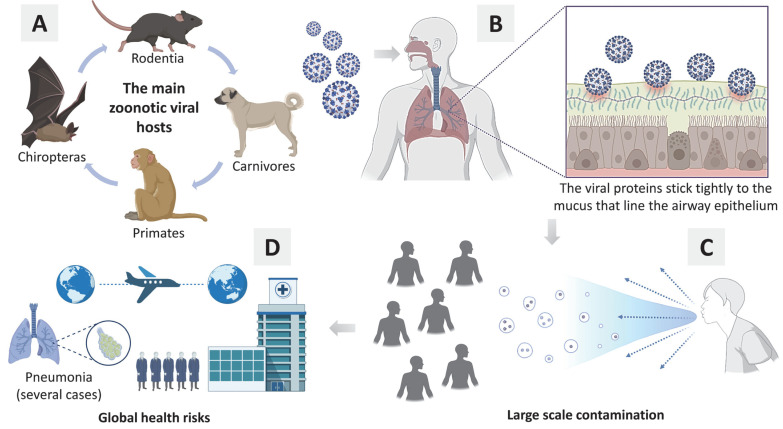
(**A**) The main animals (primary hosts) responsible for zoonotic viral transmission; (**B**) Basic general mechanism about viral infection in human cells; (**C**) Contamination of several people in society; (**D**) Global health risks (e.g., pandemic, hospital overcrowding, several cases of pneumonia, thousand deaths).

**Figure 2 molecules-27-01542-f002:**
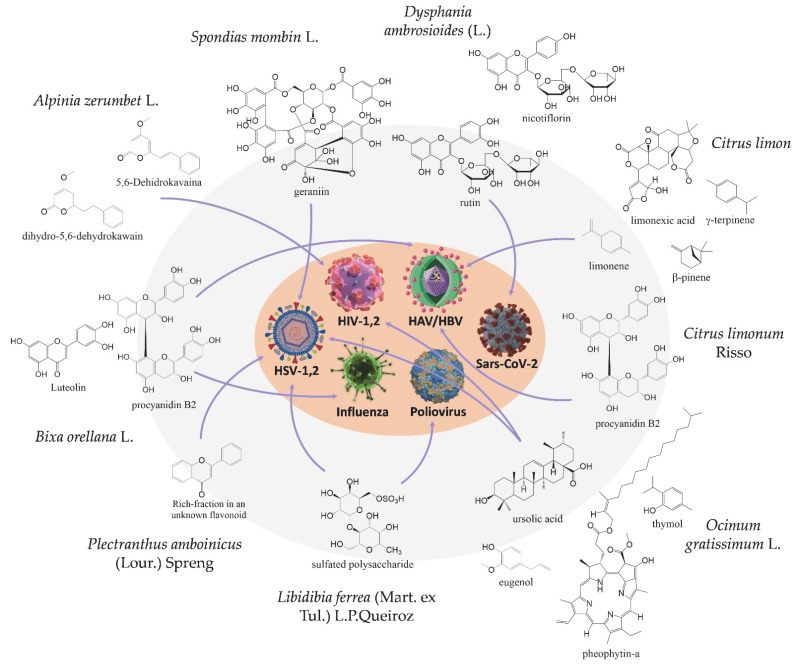
Compounds that, until this review, have already been reported to have antiviral activity for each species. Therefore, all compounds are referenced in Table 2, except *Plectranthus amboinicus,* from which a rich fraction of an unknown flavonoid was reported.

**Figure 3 molecules-27-01542-f003:**
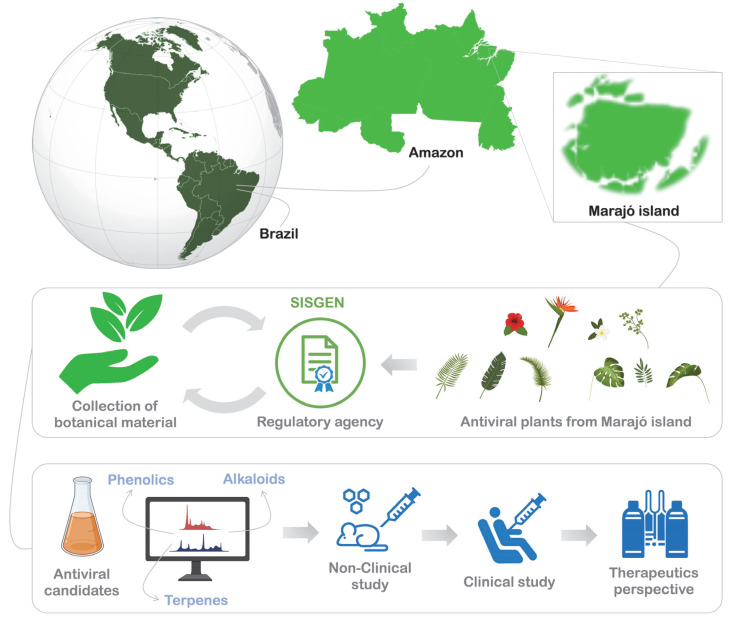
Future prospects for the use of medicinal plants from Marajó island for the discovery of antiviral candidates and the development of therapeutic alternatives.

**Table 1 molecules-27-01542-t001:** The main plants cited to treat viral diseases in Marajó Island.

Species	Occurrence *		Native Name in Marajó	Part of Plant	Use and Indication
	Continent	Region			
*Alpinia**zerumbet* L.	Southern America	Brazil	Vindicá	L, B	Infusion used to treat common viruses
*Bixa Orellana* L.	Europe, Southern America	Northern europe, Brazil	Urucum	L	Infusion used to treat flu
*Citrus* *limon*	Southern America	Brazil	Limãozinho	JF	Juice used to treat flu
*Citrus limonum* Risso	Southern America	Brazil	Limoeiro	BF	Infusion used to eliminate secretion from the lungs
*Dysphania ambrosioides* (L.)	Europe, Southern America	Northern europe, Brazil	Mastruz	L	Juice is used to eliminate secretion from the lungs in viral infections
*Libidibia ferrea* (Mart. ex Tul.) L.P.Queiroz	Southern America	Brazil	Jucá	S	Infusion used to treat flu
*Ocimum gratissimum* L.	Southern America	Brazil	Alfavacão	L	Tea used to treat flu and cough
*Plectranthus amboinicus* (Lour.) Spreng	South Africa, Southern America	Kenya, Angola, Mozambique, Swaziland, northern Natal, Brazil	Hortelã-Grande	L	Tea used to treat inflammation and sore throat
*Spondias mombin* L.	Southern America	Brazil	Taperebazeiro	L, B	Infusion used against herpes virus

Note: L: Leaves; B: Bark; BF: Bark of Fruit; S: Seeds; JF: Juice of Fruit; O: Oil; * http://www.worldfloraonline.org/ (accessed on 15 January 2022).

## Data Availability

All support data used in this study are available from the authors.

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
