# Peer review of "Antiviral Plants from Marajó Island, Brazilian Amazon: A Narrative Review"

_molecules, 2022, doi:10.3390/molecules27051542_

Round 1

Reviewer 1 Report

The current review reports the plants growing in Marajó island and showing antiviral potential against diverse viruses including HIV-1,2, HSV-1,2, Sars-CoV-2, HAV and HBV, Poliovirus, and influenzas.

Conclusions regarding classes of compounds such as cyanogenic glycosides, alkaloids, phenols, essential oils, in terms of antiviral potential.

Figure such as number of pubications published per year could give more concluding analyses.

Figure summarizing classes of compounds exhibiting antiviral potential: Phenolics, alkaloids, terpenes,

some perspectives need to be added to the conclusions part to see how the field could go on the future, how we can make use of that

minor Eglish check

Author Response

Reviewer general comments. The current review reports the plants growing in Marajó island and showing antiviral potential against diverse viruses including HIV-1,2, HSV-1,2, Sars-CoV-2, HAV and HBV, Poliovirus, and influenzas.

Authors – We appreciate your contributions, and they certainly gave us a new perspective of our manuscript. Also, our aim for this review was to show plants that are included in the primary treatment of the Marajó population, which for a long time have subsidized the demand for basic health in the region. A brief metric review, based on the systematic nomenclature of these species, allowed us to discover a very promising segment of the scientific window in which these plants are already found. In addition, on these aspects, we saw that our manuscript can be a bridge so that future investigations can be evaluated against the most common viruses that affect the population, for instance, HIV-1,2, HSV-1,2, Sars-CoV-2, HAV and HBV, Poliovirus, and influenzas.

Reviewer. Conclusions regarding classes of compounds such as cyanogenic glycosides, alkaloids, phenols, essential oils, in terms of antiviral potential.

Authors – Dear reviewer, thank you for this comment. It is an error that came from previous versions, but it was removed from this version.

Reviewer. A figure such as the number of publications published per year could give more concluding analyses.

Authors – Dear reviewer, I’m sorry but I didn’t understand your recommendation. However, if these are inherently in the index of publications to these species, we consider that they are included in the specific sections is shown the most current panorama regarding studies of each species, although pioneer for some.

Reviewer. Figure summarizing classes of compounds exhibiting antiviral potential: Phenolics, alkaloids, terpenes,

Authors – Thank you for your comment. Based on this, we created figure 3 to summarize classes of compounds exhibiting antiviral potential

Reviewer. some perspectives need to be added to the conclusions part to see how the field could go in the future, how we can make use of that

Authors – We added to section 6 for discussion about future perspectives. Also, figure 3 was created to summarize it.

Reviewer. minor English check

Authors – The language was checked and corrected.

Reviewer 2 Report

This review article summarizes the antiviral compounds discovered from the plants in Marajó island. Natural products provide an invaluable reservoir for researchers to search for promising candidates for diseases treatment. This resubmitted manuscript made some improvements compared to the previous version. The activities (IC50) data and high-quality figures were provided. The nanotechnology part was eliminated. However, as a review article, this paper still lacks a comprehensive summary of those bioactive compounds. For example:

  • The reviewer assumes that the nine compounds shown in Figure 2 represent each species, but they are not the only compound in each species. If there are more compounds to be introduced, we would like to see the structures of them. Otherwise, the information provided by this review is minimal.
  • The stereo configuration of compounds is missing, especially for the sugar moiety.
  • Some of the active compounds are detailed in their specific names, while others are labeled as “Flavonoids” or “Terpene”. The authors still mixed up the concept of class, subclass, and their representative compounds.
  • The grammar and some sentences still need to be corrected by native speakers.

Therefore, the reviewer considers that this manuscript failed to provide sufficient information to readers. Much more search and summary work have to be done to prepare a qualified review article.

Author Response

Reviewer general comments. This review article summarizes the antiviral compounds discovered from the plants in Marajó island. Natural products provide an invaluable reservoir for researchers to search for promising candidates for diseases treatment. This resubmitted manuscript made some improvements compared to the previous version. The activities (IC50) data and high-quality figures were provided. The nanotechnology part was eliminated. However, as a review article, this paper still lacks a comprehensive summary of those bioactive compounds. For example:

Reviewer. The reviewer assumes that the nine compounds shown in Figure 2 represent each species, but they are not the only compound in each species. If there are more compounds to be introduced, we would like to see their structures of them. Otherwise, the information provided by this review is minimal.

Authors – Dear, the legend of figure 1 was rewritten to be clearer. However, in figure 1 or the text, we did not say that those compounds represent each species. To be clear, those compounds are which so far, already reported with antiviral activity for each species. Of course, there are several other chemical compounds for each species, but without any antiviral properties. In this sense, these other compounds would not add to this review.

Reviewer. The stereo configuration of compounds is missing, especially for the sugar moiety.

Authors – stereochemistry was corrected to procyanidin B2, Rutin, Geraniin, and Sulfated polysaccharide.

Reviewer. Some of the active compounds are detailed in their specific names, while others are labeled as “Flavonoids” or “Terpene”. The authors still mixed up the concept of class, subclass, and their representative compounds.

Authors – It was corrected in Table 2 as well on the main text.

Reviewer. The grammar and some sentences still need to be corrected by native speakers.

Authors – The English were reviewed, and the prepositions and statements were corrected.

Reviewer. Therefore, the reviewer considers that this manuscript failed to provide sufficient information to readers. Much more search and summary work have to be done to prepare a qualified review article.

Authors – The authors reinforce that all scientific literature that is available on antiviral properties for these plants is in this review. We understand that some preliminary data are were generated from small-scale research projects to evaluate the feasibility, before conducting full research studies (for this is necessary quite grant support, several people, material, and a long time). Therefore, we believe that all scientific literature available so far, can improve and be useful to new research.

Reviewer 3 Report

molecules-1543676-peer-review

The manuscript has important disadvantages in the presentation of the information (too preliminary and lack of novelty), of which the main weakness is mentioned below:

  • The bibliography must be reviewed, there are incomplete or not citations according to the Journal. On the other hand, a review and update of the recently reported bibliography should be carried out. 

 For example reference11

  • The manuscript does not show a solid discussion, it only shows or reports previous results on the plants. This should be reviewed by the authors.

Some results incorporated into the manuscript are very weak and preliminary and are reported in non-mainstream journals. An example is that of the species Plectranthus amboinicus:

Lines 144-148, page 5: The authors write “ Flavonoids isolated from Plectranthus amboinicus… antiviral effect”

International Journal of Pharmacognosy and Phytochemical Research 2016; 8(6); 1020-1024https://www.scimagojr.com/journalsearch.php?q=20400195018&tip=sid

“The aqueous extract of Plectranthus amboinicus showed 63% inhibition of HIV –1 protease at a concentration of 100 µg/ml. Acetyl pepstatin was taken as positive control and had a 58% inhibition at 30µg/ml. Figure 2a-b depicts representative HPLC pattern of peptide substrate during different test conditions

Flavonoids were the major components of PA- 9 fraction of Plectranthus amboinicus.Table 3 depicts the result of the phytochemical tests carried out on PA-9

Table 3: Phytochemical analysis of bioactive fraction of PA-9”

However, in the referred manuscript, from which the corresponding paragraph is transcribed, the authors do not isolate any flavonoids, they only carry out preliminary phytochemical tests, through qualitative reactions, and they take for granted that the extract or fraction is positive for flavonoids. This is very preliminary and not relevant.

  • Please review and update the references for each species, there are recent reports on chemical composition .

Cheng, C.Y., Kao, C.L., Li, H.T. et al. A New Flavonoid from Plectranthus amboinicus. Chem Nat Compd 57, 30–32 (2021). https://doi.org/10.1007/s10600-021-03274-5

Pharmacognosy Journal,2020,12,6s,1573-1577. DOI:10.5530/pj.2020.12.215   On the other hand, in Figure 2, a figure of a typical flavonoid would be more appropriate supported by previous report.
  • The authors report in Table 1 the local uses of the species, however, some species grow or come from other parts of the world, where they have other uses in traditional medicine, which should be mentioned with their respective bibliography.

An example is that of the species Plectranthus amboinicus (Lour.) Spreng.

http://www.worldfloraonline.org/taxon/wfo-0000275221

  • For each species, its geographical distribution should be incorporated in an additional column in Table 1, please use for this http://www.worldfloraonline.org/

  • Studies on antiviral activity are too preliminary and of little interest to potential journal readers if they are not accompanied by reference parameters (relevant literature) and reference compounds that allow quantifying the potential of an extract or compound as antiviral. Authors should include, analyze and discuss in the discussion section that is suggested  bibliographic references that they have considered to evaluate the potential of the reported plants. 

As an example of what is requested, the following bibliography is mentioned that gives parameters for antimicrobial activity

Plese see reference Kuete and Efferth 2010, where antibacterial activity parameters are given for extracts and pure compounds. Extract: significant (MIC<100μg/ml), moderate (100<CMI≤625μg/ml) or weak (CMI>625 μg/ml).

For compounds, this stringent endpoints criteria were: significant (MIC<10μg/ml), moderate (10<MIC≤100 μg/ml) and low or negligible (MIC > 100 μg/ml) Kuete V, Efferth T (2010) Cameroonian medicinal plants: pharmacology and derived natural products. Front Pharmacol 1:123"

Please see Journal of Ethnopharmacology Volume 100, Issues 1–2, 22 August 2005, Pages 80-84. Perspective paper Medicinal plants and antimicrobial activity J.L.RíosM.C.Reciohttps://doi.org/10.1016/j.jep.2005.04.025

“whereas the presence of activity is very interesting in the case of concentrations below 100 µg/ml for extracts and 10µg/ml for isolated compounds”.

In its current state the manuscript is not suitable for the Journal (lack of novelty and too preliminary)

Author Response

Reviewer general comments. The manuscript has important disadvantages in the presentation of the information (too preliminary and lack of novelty), of which the main weakness is mentioned below:

Reviewer. The bibliography must be reviewed, there are incomplete or no citations according to the Journal. On the other hand, a review and update of the recently reported bibliography should be carried out. For example, reference 25

Authors - We appreciate your comment. Some corrections were performed on the main text. Regarding reference 25, several reviews were made, and there is no recently reported antiviral bibliography other than reference 25.

Reviewer. The manuscript does not show a solid discussion, it only shows or reports previous results on the plants. This should be reviewed by the authors.

Authors – The authors are thankful for your comments. The corrections were performed on the main text.

Reviewer. Some results incorporated into the manuscript are very weak and preliminary and are reported in non-mainstream journals. An example is that of the species Plectranthus amboinicus: Lines 144-148, page 5: The authors write “Flavonoids isolated from Plectranthus amboinicus… antiviral effect”

International Journal of Pharmacognosy and Phytochemical Research 2016; 8(6); 1020-1024https://www.scimagojr.com/journalsearch.php?q=20400195018&tip=sid

Authors – The authors agree that the results are preliminary yet. However, it is a pioneer study regarding antiviral properties for that species and can be a guide to start advanced research shortly. Furthermore, this manuscript was recently cited by high mainstream journals, for example in the Frontiers in Molecular Biosciences (https://doi.org/10.3389/fmolb.2020.613401), Journal of Infection and Public Health (https://doi.org/10.1016/j.jiph.2020.09.002), Scientific Reports, Nature (https://doi.org/10.1038/s41598-018-22485-5). Thus, the manuscript from Plectranthus amboinicus is an excellent contribution and super useful to science development.

Reviewer. “The aqueous extract of Plectranthus amboinicus showed 63% inhibition of HIV –1 protease at a concentration of 100 µg/ml. Acetyl pepstatin was taken as positive control and had a 58% inhibition at 30µg/ml. Figure 2a-b depicts a representative HPLC pattern of peptide substrate during different test conditions

Flavonoids were the major components of the PA- 9 fractions of Plectranthus amboinicus. Table 3 depicts the result of the phytochemical tests carried out on PA-9 Table 3: Phytochemical analysis of a bioactive fraction of PA-9”

However, in the referred manuscript, from which the corresponding paragraph is transcribed, the authors do not isolate any flavonoids, they only carry out preliminary phytochemical tests, through qualitative reactions, and they take for granted that the extract or fraction is positive for flavonoids. This is very preliminary and not relevant.

Authors – Thank you for the correction. This statement was corrected to “A flavonoid-rich fraction of the Plectranthus amboinicus showed antiviral activity against HIV-1 the Herpes Simplex Virus type 1”. In addition, the authors do not appreciate as a data not relevant, as already reported in a journal of the prestigious nature group, those experiments are reliable qualitative examination for the detection of various phytochemical constituents (Scientific Reports, https://doi.org/10.1038/s41598-020-74262-y)

Reviewer. Please review and update the references for each species, there are recent reports on chemical composition.

Cheng, C.Y., Kao, C.L., Li, H.T. et al. A New Flavonoid from Plectranthus amboinicus. Chem Nat Compd 57, 30–32 (2021). https://doi.org/10.1007/s10600-021-03274-5

Pharmacognosy Journal,2020,12,6s,1573-1577. DOI:10.5530/pj.2020.12.215   On the other hand, in Figure 2, a figure of a typical flavonoid would be more appropriately supported by the previous report.

Authors - Dear reviewer, we appreciate your suggestions. However, our review is not describing the chemical composition of any species. Furthermore, the papers suggested not showing any antiviral flavonoid or any other compounds. For instance, the manuscript DOI: 10.1007/s10600-021-03274-5 showed a new flavonoid isolated from Plectranthus amboinicus, but without any biological activity. The second manuscript (DOI:10.5530/pj.2020.12.215) related anticancer activity and in vitro antioxidant of flavonoids in Plectranthus amboinicus, and anticancer and antioxidant compounds are not our objective in this review about antiviral plants from Marajó Island. For example, table 1 reports flavonoids because it has been related to antiviral activity in the literature (Reference 25). Therefore, all compounds described in table 2 were reported previously with antiviral activity.

Reviewer. The authors report in Table 1 the local uses of the species, however, some species grow or come from other parts of the world, where they have other uses in traditional medicine, which should be mentioned with their respective bibliography. An example is that of the species Plectranthus amboinicus (Lour.) Spreng.

http://www.worldfloraonline.org/taxon/wfo-0000275221

For each species, its geographical distribution should be incorporated in an additional column in Table 1, please use for this http://www.worldfloraonline.org/

AuthorsDear reviewer, a column was added in Table 1 with occurrence, continent, and region. Thank you!

Reviewer. Studies on antiviral activity are too preliminary and of little interest to potential journal readers if they are not accompanied by reference parameters (relevant literature) and reference compounds that allow quantifying the potential of an extract or compound as antiviral. Authors should include, analyze and discuss in the discussion section that is suggested bibliographic references that they have considered to evaluate the potential of the reported plants. 

As an example of what is requested, the following bibliography is mentioned that gives parameters for antimicrobial activity

Please see reference Kuete and Efferth 2010, where antibacterial activity parameters are given for extracts and pure compounds. Extract: significant (MIC<100μg/ml), moderate (100<CMI≤625μg/ml) or weak (CMI>625 μg/ml).

For compounds, this stringent endpoints criteria were: significant (MIC<10μg/ml), moderate (10<MIC≤100 μg/ml) and low or negligible (MIC > 100 μg/ml) Kuete V, Efferth T (2010) Cameroonian medicinal plants: pharmacology and derived natural products. Front Pharmacol 1:123"

Please see Journal of Ethnopharmacology Volume 100, Issues 1–2, 22 August 2005, Pages 80-84. Perspective paper Medicinal plants and antimicrobial activity J.L.RíosM.C.Reciohttps://doi.org/10.1016/j.jep.2005.04.025

“whereas the presence of activity is very interesting in the case of concentrations below 100 µg/ml for extracts and 10µg/ml for isolated compounds”.

Authors – Dear reviewer, thank you for your comments. So far, there is not any relationship between antimicrobial activity (estimated with a MIC or MBC) and antiviral activity (estimated with TCID50/mL or IC50), such as the action mechanism. Based on this, we do not consider it valid to use antimicrobial information to ranking antiviral data without any scientific evidence.

Reviewer. In its current state, the manuscript is not suitable for the Journal (lack of novelty and is too preliminary)

Authors – Dear reviewer, thank you very much for your valuable contribution to the improvement of our manuscript. We carefully evaluate your recommendations and made the indicated corrections, which we highlighted in change control in the new version submitted for your conference. The suggestions and comments that the authors do not agree with were supported by literature data. We also took the opportunity to re-write and check other points that are included in the new version submitted.

Round 2

Reviewer 2 Report

This revised manuscript is capable to be published on the journal Molecules. 

Author Response

Dear reviewer, we greatly appreciate your contributions to improve the
presentation of our work.

Reviewer 3 Report

Molecules-1543676- second review

The authors have made some changes to the manuscript, however some suggestions have not been considered, I apologize because perhaps I have not been clear in the writing of them.  After minor suggestions, the manuscript should be accepted for publication

  • Reference 26 is incomplete or not citations according to the Journal.
  1. Thayil, S.; Thyagarajan PA-9 : A Flavonoid Extracted from Plectranthus amboinicus Inhibits HIV-1 Protease. 2016.

         Please include Journal name 

 International Journal of Pharmacognosy and Phytochemical Research 2016; 8(6); 1020-1024

  • Some results incorporated into the manuscript are very weak and preliminary and are reported in non-mainstream journals. An example is that of the species Plectranthus amboinicus: Lines 144-148, page 5: The authors write “Flavonoids isolated from Plectranthus amboinicus… antiviral effect”

International Journal of Pharmacognosy and Phytochemical Research 2016; 8(6); 1020-1024https://www.scimagojr.com/journalsearch.php?q=20400195018&tip=sid

Authors – The authors agree that the results are preliminary yet. However, it is a pioneer study regarding antiviral properties …….Thus, the manuscript from Plectranthus amboinicus is an excellent contribution and super useful to science development.

The observation or suggestion made is aimed at the fact that the authors mention isolated flavonoids and in the reference 26 (Thayil, S.et al, 2016)) there is no isolation, only a preliminary screening of chemical compounds. Now the text as presented is more appropriate.

  • Studies on antiviral activity are too preliminary and of little interest to potential journal readers if they are not accompanied by reference parameters (relevant literature) and reference compounds that allow quantifying the potential of an extract or compound as antiviral. Authors should include, analyze and discuss in the discussion section that is suggested bibliographic references that they have considered to evaluate the potential of the reported plants.

Authors – Dear reviewer, thank you for your comments. So far, there is not any relationship between antimicrobial activity (estimated with a MIC or MBC) and antiviral activity (estimated with TCID50/mL or IC50), such as the action mechanism. Based on this, we do not consider it valid to use antimicrobial information to ranking antiviral data without any scientific evidence.

Dear authors. I apologize for the suggestion, which has not been interpreted, probably due to the wording of it.

I have only suggested, if possible, that a bibliographic reference could be incorporated that mentions parameters or ranges (TCID50/mL or IC50) to consider an extract or compound with high, moderate, or low antiviral activity.

As an example I have mentioned and included some references for the case of antimicrobial activity (MIC, MBC) of extracts or compounds. Which I have not requested to be incorporated into the manuscript, nor used as parameters.

Author Response

Dear reviewer, thank you so much for your comments.

Reference 26 was corrected as well as others.

About the value of reference for antiviral drugs. We apologize also for not understanding your previous suggestion. However, so far, we could not find a reference in the literature for a high, moderate or low activity, as it is usually related to the selectivity index (drug toxicity), and a drug can have a high IC50 but be toxic. 

Again, we are grateful for you review process.